# Soil Organic Carbon Distribution and Its Response to Soil Erosion Based on EEM-PARAFAC and Stable Carbon Isotope, a Field Study in the Rocky Desertification Control of South China Karst

**DOI:** 10.3390/ijerph19063210

**Published:** 2022-03-09

**Authors:** Xinwen Wang, Ziqi Liu, Kangning Xiong, Yuan Li, Kun Cheng

**Affiliations:** 1School of Karst Science, Guizhou Normal University, Guiyang 550001, China; wangxinwen97@126.com (X.W.); liyuan7pro@163.com (Y.L.); chengkun99598@163.com (K.C.); 2State Engineering Technology Institute for Karst Desertification Control, Guiyang 550001, China

**Keywords:** soil organic carbon, vegetation restoration, soil erosion, dissolved organic matter, stable carbon isotope, karst

## Abstract

Ecological restoration plays an important role in enhancing carbon sequestration ability in karst areas, and soil organic matter is one of the main carbon reservoirs in karst key zones. The serious soil erosion in karst areas leads to the loss of soil organic matter (SOM). However, the distribution characteristics of SOM and its response mechanism to soil erosion in the process of rocky desertification control have rarely been reported. In this study, soil samples of five restoration types (abandoned land, AL; grassland, GL; peanut cultivated land, PCL; *Zanthoxylum bungeanum* land, ZBL; forest, FS) were collected in typical karst rocky desertification drainage, south China. By measuring soil organic carbon (SOC), dissolved organic carbon (DOC), and δ^13^C_soc_ values and combining with spectral tools, the distribution and isotopic composition of soil shallow organic carbon in definitized karst drainage was definitized and the response of DOM spectral characteristics to soil erosion was explored. The results showed that three kinds of fluorescence components were detected by fluorescence excitation emission matrix (EEM)-parallel factor analysis (PARAFAC), C1 and C2 were humus-like, and C2 was protein-like. Abandoned could be a more suitable control measure for enhancing SOC quality in the karst regions of south China. The variation trend of SOC content, δ^13^C_soc_ values, spectral indexes, and the distribution of fluorescence components from the midstream to downstream of the drainage indicated the soil redistribution. This study provides basic scientific data for karst rocky desertification control and for enhancing the soil carbon sequestration ability of karst.

## 1. Introduction

Soil organic carbon pool is the largest carbon pool in terrestrial ecosystems and an important part of the global carbon cycle [1,2]. Improving soil carbon sequestration ability is an important aspect of achieving carbon peak and carbon neutrality in China [3]. Dissolved organic matter (DOM) is widely found in soil [4], and is the most active and bioavailable component in soil organic matter (SOM), with strong reactivity and migration ability [5]. DOM can combine with soil particles and coexist with minerals, thus becoming part of the organic carbon pool [6]. DOM plays an important role in the biogeochemical cycle of carbon and other elements by influencing the adsorption, dynamic displacement, residence time, and microbial availability of soil organic matter, and it is the core of the study of carbon sequestration and nutrient availability in soil [1,7]. DOM characteristics are closely related to biogeochemical processes involved in periodic changes of soil [8,9], and soil erosion is an important factor leading to spatial changes of DOM [10]. The process of soil erosion generally includes four stages: separation, decomposition, migration, and deposition [1]. These stages will not only lead to changes in the spatial distribution of soil organic matter content, but will also lead to changes in the properties of soil organic matter [7,10].

UV–visible absorption spectrum and fluorescence spectrum are widely used optical tools to obtain the source, composition, and activity of organic matter in soil or water environment [10,11]. Stable carbon isotope can be used as a marker to indicate the source, transformation, and release of soil organic carbon [12,13]. Studies by Yang et al. [14] showed that fluorescence EEM-PARAFAC was effective in discriminating DOM sources. He et al. [15] compared the spectral characteristics of alkaline extractable organic matter in pore water and river sediments to understand the distribution behavior of sediment organic matter between dissolved phase and granular phase. Che et al. [16] studied the turnover of organic carbon in frozen soil by using stable carbon isotope technology. Spectral analysis and stable carbon isotope technology can effectively distinguish the distribution and migration process of carbon elements.

Carbon sinks in karst regions around the world account for more than one-third of the global “missing sinks” [17,18]. According to zoning calculation, carbon sinks in karst regions in China amount to 36,991 million tons [19], making it significant for improving ecological carbon sink capacity [20]. The soil habitat in karst areas is fragile and susceptible to erosion. Karst areas in China are mainly distributed in Yunnan, Guangxi, and Guizhou provinces. Since the ninth Five-Year Plan, focusing on the problems of karst ecological restoration and rocky desertification control, systematic theoretical studies have been carried out on karst ecological restoration and rocky desertification control in karst areas in south China, and a series of comprehensive biological and engineering measures have been taken to solve the problems of rocky desertification and soil erosion [21]. Previous studies on karst ecological restoration mostly focused on the development of economic vegetation restoration models [22,23], soil physical and chemical properties of different vegetation types used for restoration [24], and soil fertility [25,26]. Although some studies have proved that vegetation restoration significantly improved SOC in karst rocky desertification areas in south China, and that restoration of abandoned land has advantages in karst areas [24], the study of soil carbon turnover in karst regions still needs to be advanced. This study focused on the different vegetation restoration soils in typical karst rocky desertification drainage; specifically, this study definitized the distribution and isotopic composition of soil organic carbon in abandoned land (AL), grassland (GL), peanut cultivated land (PCL), *Zanthoxylum bungeanum* land (ZBL), and forest (FS), and explored the effects of vegetation restoration and soil erosion on the distribution of soil organic matter. We hypothesize that (1) the effect of abandoned land on SOC conservation is better than that of restoration land with other vegetation and (2) there is an obvious redistribution of soil organic carbon from upstream to downstream.

## 2. Materials and Methods

### 2.1. Study Area 

The karst area centered on Guizhou Plateau in south China is the largest and most concentrated karst ecologically fragile area in the world, covering an area of more than 55 × 10^4^ km^2^ [22]. It is also the most typical karst development, the most complex, and the most abundant in landscape types. Rock desertification is the most serious ecological and environmental problem facing this region [22]. We selected the representative Chaeryan drainage as the research area in the Guizhou plateau mountainous area, which represents the overall structure of the karst environment in South China (Figure 1). The local climate is a south subtropical, dry, hot valley climate with an average annual precipitation of 1100 mm and an average annual temperature of 18.4 °C [27]. The drainage has an elevation range of 607–890 m, presenting a terrain pattern of high in the south and low in the northeast. Heavy rain is frequent in the wet season, and two discontinuous main ephemeral streams and one confluence point are formed in the drainage due to rainfall events, while the flow is cut off in the calm and dry seasons. The soil distribution in the region is highly heterogeneous and barren, mainly yellow soil and yellow lime soil with an average thickness of 25 ± 10 cm and a rock outcrop ratio of 90%, and the soil is seriously eroded [27,28]. The vegetation is mainly broadleaved forest, and coniferous and broadleaved mixed forest and shrub. The original vegetation is seriously damaged, and now it is mainly secondary vegetation. The main land use types are agricultural farmland, abandoned land, forestland, etc. For ecological restoration of rocky desertification and regional economic development, *Zanthoxyhum planispinum var. Dingtanensis* is planted in a large number in the drainage area.

### 2.2. Sample Collection and Preparation

#### 2.2.1. Sample Collection

According to the terrain distribution characteristics of the drainage, a total of 14 sampling points were set up in the upper reaches (A1), middle reaches (A2), and lower reaches (A3) of the drainage during the wet season in August 2020. The sample points were classified by ground potential difference (site) and vegetation cover type. The basic information of the sample points is shown in Table 1. The types of vegetation restoration included abandoned land (AL), grassland (GL), peanut cultivated land (PCL), *Zanthoxylum bungeanum* land (ZBL), and forest (FS). Two soil layers of 0–15 cm and 15–30 cm were collected, and three parallel samples were collected. 

#### 2.2.2. Laboratory Analysis

Carbon stable isotope determination: Soil samples (5 g) were weighed with a 60-mesh sieve, soaked in 1 mol/L hydrochloric acid solution for 24 h to remove the carbonates in the soil, washed with deionized water to neutral, dried at 60 °C, and then ground. Samples of 2–5 mg were weighed with a 1/10,000 balance (WXTS3DU, Mettler Toledo, Zurich, Switzerland) and wrapped in tin cups. The δ^13^Csoc of the samples was determined by elemental analysis–stable isotope ratio mass spectrometry (EA IsoLink & Delta V Advantage, Thermo Fisher, Waltham, MA, USA). Test accuracy was ≤0.1‰

Preparation of DOM solution: The air-dried soil sample was ground through a 60-mesh screen. A certain amount of screened and air-dried soil was weighed, and ultra-pure water was added at the soil–water ratio of 1:5. After shaking and centrifugation, the supernatant was filtered through 0.22 μm polyethersulfone drainage membrane to obtain soil DOM solution. The DOC content of all samples was diluted to less than 10 mg/L. A TOC analyzer (multi N/C 3100, Analytik jena, Jena, Germany) was used to determine DOC, ensuring that the UV–visible absorbance at 254 nm was less than 0.3, so as to reduce the internal filtration effect of subsequent fluorescence scanning [29].

Spectral scanning: UV–visible absorption spectra were scanned using a UV–visible absorption spectrometer (SPECORD Plus 200, Analytik Jena, Jena, Germany), with ultra-pure water (CascadaII, I20, Pall, New York, NY, USA) blank. In the range of 200~800 nm, quartz cuvettes with optical path of 10 mm were scanned at intervals of 1 nm. A fluorescence spectrometer (RF-5301PC, Shimadzu, Japan) was used for 3D fluorescence spectra. The Ex scanning range was 220–500 nm, the Em scanning range was 250–600 nm, the Ex interval was 5 nm, the Em interval was 1 nm, and the scanning speed was 2400 nm/min. Ultra-pure water (CascadaII, I20, Pall, New York, NY, USA) blank was deducted to remove scattering [30].

Determination of SOC: Soil samples (5 g) were weighed with a 60-mesh sieve, soaked in 1 mol·L^−1^ hydrochloric acid solution for 24 h to remove the carbonates in the soil, washed with deionized water to neutral, dried at 60 °C, and then ground. Samples of 20–50 mg were weighed with a 1/10,000 balance and wrapped in tin cups. The SOC of the samples was determined by an elemental analyzer (FlashSmart, Thermo Fisher, Waltham, MA USA). For specific methods, please refer to *Determination of soil organic carbon by combustion oxidation—non-dispersive infrared method* (HJ 695-2014).

### 2.3. PARAFAC

Parallel factor analysis (PARAFAC) was used to analyze EEMs and identify the number, type, and intensity of CDOM fluorophore. PARAFAC consensus identified 84 EEMs (2 layers, 14 samples, and 3 replicates), and DOMFluor toolbox was used to analyze the matrix group. Component models (2–7) with non-negative constraints were used in the PARAFAC analysis. Residual analysis, dichotomy analysis, random initialization, and visual detection were used to determine the number of fluorescent components [31]. During the analysis, four samples were removed as outliers. The relative abundance of each component was reflected by the maximum fluorescence intensity (Fmax) per unit DOC (Fmax/DOC), and the percentage of each PARAFAC component in the total fluorescence was calculated as the Fmax value of each component divided by the sum of the Fmax of each component.

### 2.4. δ^13^C_soc_ and Spectral Index 

Formula for calculating δ^13^C_soc_ isotope ratio of samples: (1)δ13C=[(C13/C12)sample/(C13/C12)standard]×1000‰
the reference standard material uses V-PDB (δ^13^C = 1.124‰) and USGS40 (δ^13^C = –26.39 ± 0.04‰).

Absorption coefficient calculation formula:(2)a(λ)=2.303A(λ)/r
where *a*(λ) is the absorption coefficient at wavelength λ, and r is the optical path length (r = 0.01 m). The ratio of absorbance at 250 nm to that at 365 nm is defined as E2:E3, which is inversely proportional to the average molecular weight of DOM. SUVA 254 is an indicator of aromaticity, calculated by a(254)/DOC [L·mg /(C·m)], which is positively correlated with the aromaticity and hydrophobicity of DOM. SUVA 260 represents the content of hydrophobic components, is a(260)/DOC [L·mg/(C·m)], and is proportional to the content of hydrophobic fraction [4,5,32,33,34].

The fluorescence index (FI) is the ratio of fluorescence intensity at 470 nm and 520 nm when the excitation wavelength is 370 nm and is widely used to distinguish between exogenous DOM and microbial DOM (FI > 1.9, microbial; FI < 1.4, exogenous). The humification index (HIX) is the ratio of the integral values of fluorescence intensity between 435–480 nm and 300–345 nm when the excitation wavelength is 254 nm, and indicates the humus content or degree of humification. HIX increases with the enhancement of humic characteristics. The biological index (BIX) is the ratio of excitation wave length of 310 nm and emission wavelength of fluorescence intensity at 380 nm and 430 nm, representing the relative contribution of authigenes (BIX > 1, autochthonous; BIX < 1, allochthonous) [35,36,37,38,39].

### 2.5. Statistical Analysis

Statistical analysis was performed using IMB SPSS Statistics 22 (IBM Inc., Chicago, IL, USA). Differences in SOC content, DOC content, and spectral indexes of δ^13^C_soc_ values between site and depth were assessed by one-way ANOVA and least significant difference (LSD) test at 95% confidence level. *p* < 0.05 was significant.

## 3. Results and Analysis

### 3.1. SOC and δ^13^C_soc_ Distribution Characteristics of Different Vegetation Restoration Types and Sites

In the 0–15 cm soil layer, the average SOC content was FS > ZBL > AL > PCL > GL, the average DOC content was FS > GL > PCL > ZBL > AL. In the 15–20 cm soil layer, the average SOC content was FS > ZBL > AL > PCL > GL, the average DOC content was PCL > FS > GL > AL > ZBL. Those showed that the SOC and DOC values of soil covered by forest were higher, while the SOC and DOC values of soil abandoned were lower (Figure 2). In terms of spatial distribution, SOC content showed a distribution characteristic of A1 > A2 > A3. In the 0–15 cm soil layer, DOC content at site A3 was 24.19% higher than that at site A1 and 52.43% higher than that at site A2 (*p* < 0.05), and there was no significant difference in DOC content in the 15–30 cm soil layer (*p* > 0.05) (Table 2). The SOC content and DOC content in the 0–15 cm soil layer were higher than those in the 15–30 cm soil layer (Table 2). In addition, soil δ^13^C_soc_ values varied from −24.18‰ to 19.51‰ (Table 2), showing the distribution characteristics of A2 > A1 > A3, indicating that the vegetation overlying the soil was mainly C3 plants [40].

### 3.2. Spectral Parameter Characteristic

#### 3.2.1. Characteristics of UV–Visible Absorption Spectrum

Soil SUVA254 values ranged from 0.41 to 1.62 L·mg/(C·m), with an average value of 0.89 ± 0.03 L·mg/(C·m). There was no significant difference in soil SUVA254 values in different plants. The values of SUVA260 ranged from 0.33 to 1.31 L·mg/(C·m), with an average value of 0.75 ± 0.03 L·mg/(C·m). The distribution of SUVA260 values was similar to that of SUVA254 values (Figure 3), compared with other vegetation restoration types, the difference between the 0–15 cm soil layer and the 15–30 cm soil layer was greater when covered by forest. E2:E3 values ranged from 3.11 to 9.45 L·mg/(C·m), with an average of 4.88 ± 0.13 L·mg/(C·m). As can be seen from UV–visible absorption indexes, the 0–15 cm soil layer covered by forest had higher aromaticity, hydrophobicity, and molecular weight compared with the 15–30 cm soil layer, while the 15–30 cm soil layer had more stable aromaticity, hydrophobicity, and molecular weight. In addition, the 15–30 cm soil layer had slightly higher aromaticity and hydrophobicity compared with the 0–15 cm soil layer.

The values of SUVA254 and SUVA260 did not change significantly at sites A1, A2, and A3 (Figure 4). E2:E3 of site A1 at 0–15 cm was significantly higher than that of site A2 and A3, indicating its low aromaticity and low molecular weight.

#### 3.2.2. Characteristics of Fluorescence

FI values of soil samples ranged from 1.54 to 2.40, with an average of 1.75 ± 0.02, indicating that the soil contained microbial DOM sources [36]. FI values of all samples were less than 1.90 and greater than 1.40 (Figure 5 and Figure 6), and values did not vary much between the two depths, nor did they change with the site. It showed that the upper, middle, and lower reaches of the soil had similar microbial sources and exogenous sources [37,38]. HIX values ranged from 0.75 to 2.88, with an average of 1.80 ± 0.07. In the 0–15 cm soil layer, HIX was 0.2 higher in A2 than in A1. In the 15–30 cm soil layer, HIX was 0.38 higher in A2 than in A1 (Figure 6). These showed that A1 soil had a higher humification degree and contained more aromatic substances [39]. The change from A2 to A3 was reversed from A1 to A2, with A3 having the highest degree of humification. BIX values were mostly below 0.8 (Figure 6), indicating that the soil microbial activity was weak [39]. BIX in the 0–15 cm soil layer was lower than that in 15–30 cm soil layer, and the closer the samples were to A3, the more obvious the difference between the two layers was, indicating that the exogenous soil in the 0–15 cm soil layer was stronger, and the endogenous behavior in the 15–30 cm soil layer was stronger moving from A1 to A3 [39].

### 3.3. PARAFAC Component Analysis of Fluorescent Substances

EEM-PARAFAC was used to identify soil sample DOM fluorescence components in the drainage, and three fluorescence components were obtained (Figure 7). After comparing with C1(Ex/Em = 320/434), C2 (Ex/Em = 290(360)/485) can be classified as exogenous humic-like [7,41], C3 (Ex/Em = 280/318) can be classified as endogenous protein-like [42]. Each fluorescence component and its quantitative analysis is shown in Figure 8. The relative abundance of C3 component at three sites and two depths was the highest, accounting for more than 35%, while the fluorescence component at A3 site was the lowest, and it decreased by 52% compared to A2.

## 4. Discussion

### 4.1. Spectral Characteristics and Isotopic Composition of Soils of Different Vegetation Restoration Types

Generally speaking, the shallow soil organic matter mainly comes from the direct input of plant litters, and the overlying vegetation types directly affect the SOC content [43,44,45,46]. DOC is a more active part in SOC, and it is closely related to microorganisms, temperature, humidity, and other factors [47]. In this study, we controlled sampling in the same period, the hydrothermal conditions in the region are relatively uniform [27], and microbial community structure and activity can be consistent. However, the distribution of land use types in the drainages is broken, and soil covered by different vegetation has frequent material exchange activities; these may lead to the DOC content not being significantly different between land use types.

The δ^13^C_soc_ and spectral indices of the five vegetation restoration types were not significantly different. The mean values of FI, HIX, and BIX were 1.75 ± 0.02, 1.80 ± 0.07, and 0.71 ± 0.01, respectively. There were no significant differences in fluorescence indexes between different vegetation restoration types (Figure 5), possibly due to the growth of vascular plants generating the same components, especially the aromatic material that can remain unchanged in the biodegradation process [48]. In comparison, the FI and BIX values of soil covered by forest were the highest, which reflected the highest microbial activity and autotrophic productivity in the soil [36,40]. The δ^13^C_soc_ values of abandoned soil were more positive, and the enrichment of organic carbon was more obvious. The distribution of SUVA260 was similar to that of SUVA254, and it was the highest in abandoned land. The spectral index of abandoned land was more stable, and the soil had stronger aromaticity and higher molecular weight. According to the field investigation, the vegetation coverage of abandoned land was high (Table 1) and the vegetation diversity was high, indicating that natural restoration and abandonment play a positive role in restoring the ecological function of soil. In addition, as the dominant species in rock desertification control, *Zanthoxyhum planispinum var. Dingtanensis* made a great contribution in rock desertification control [18,23]. It can be seen from this study that aromaticity, hydrophobicity, and molecular weight of different soil layers planted with *Zanthoxyhum planispinum var. Dingtanensis* were relatively stable, and the soil in the 15–30 layer had higher aromaticity, hydrophobicity, and molecular weight, and the positive effect of *Zanthoxyhum planispinum var. Dingtanensis* on soil was basically the same as that of the abandonment restoration method. 

This study confirmed the positive effect of *Zanthoxyhum planispinum var. Dingtanensis* on karst soil restoration. *Zanthoxyhum planispinum var. Dingtanensis* forest not only plays a role in soil fixation, but also is an important economic forest in this environment [23]. Therefore, from the perspective of management, expanding the planting area of economic forest can promote the balanced development of local economy and environmental protection.

### 4.2. Effects of Soil Erosion on the Distribution of Soil Organic Matter in the Drainage

As mentioned above, the value of SUVA254 is directly proportional to the soil aromaticity, and the higher the aromaticity is, the more stable the soil organic matter is. In this study, the mean value of SUVA254 in the drainage was 0.89 ± 0.03, and it was significantly lower than that in the Loess Hilly drainage (the mean value of fallow farmland was the lowest: 4.86 ± 0.47) [7]; however, it was close to the periodically washed water-level fluctuation zone of the Three Gorges (mean: 0.36 ± 0.27) [49]. Due to the scouring of periodic floods, the soil and sediment organic matter in the water-level fluctuation zone will be released into the water [50]. Under the same conditions of plant litters input, the SUVA254 value was 5.46 times higher than that in the rocky desertification drainage, and annual rainfall (542.5 mm) [7] was 2.03 times higher in the Rocky desertification drainage (1100 mm) than in the Loess Hilly area, indicating that soil erosion by rainfall was significantly more serious in the rocky desertification drainage than in other areas.

In this study, SUVA254 and BIX values showed a downward trend from site A2 to site A3, showing that there was less aromaticity in site A3 and it was more exogenous. Moreover, Fmax/DOC of each component at sites A2 to A3 showed a decreasing trend, and the endogenous component (C3) decreased by 52% at A3 compared to that at A2, while DOC content increased. The above results are similar to the research results of Zhang et al. [10] and Liu et al. [7], which showed the change trend of soil from A2 erosion to A3 deposition. In recent studies, Zhang et al. [10] believed that soil erosion breaks the surface soil aggregates and other structures, and unstable carbon is deposited at a lower place with soil particles subjected to runoff, soil midflow, gravity, and other factors. However, the change trend of each index from A1 to A2 showed an opposite pattern; most of *Zanthoxyhum planispinum var. Dingtanensis* land is concentrated in the middle reaches of drainage, and the effect of soil and water conservation of *Zanthoxyhum planispinum var. Dingtanensis* may be the reason why there was no continuous erosion and deposition pattern in A1–A2–A3. As described in the results and analysis section, from A1 to A2 and then to A3, the soil had similar microbial and endogenous characteristics, and the microbial activity was generally low, the variation trend of δ^13^C_soc_ values from upstream to downstream was the same as that of BIX, both of which increased first and then decreased (Figure 3 and Figure 7). Therefore, A1 and A3 have stronger microbial activity than does A2. In the drainage area, the SOC and DOC contents in the 15–30 cm soil layer were lower than those in the 0–15 cm soil layer, while there was no significant difference in the DOC content in the 15–30 cm soil layer (Table 2). Moreover, the exogenous characteristics of the 0–15 cm soil layer were more obvious, indicating that the degree of soil erosion redistribution in the 0–15 cm soil layer was more obvious, and the unstable carbon in the 0–15 cm soil layer was easier to release. Therefore, it can be considered that planting plants with developed and short roots can help to preserve surface soil organic carbon. In addition, three fluorescent components were detected, C1 and C2 are exogenous humus-like, and C3 is endogenous protein-like. All soil samples showed the highest relative abundance of C3 component (Figure 8), and it is consistent with the previous analysis results of the HIX value (Figure 6), that is, the DOM humus degree of the whole soil in the drainage was weak and there was an important recent natural source. Moreover, the components were similar to those detected by Yao Xin et al. [51] in a karst water system, which reflected that soil organic matter in karst areas has a strong release effect to groundwater and surface water systems under the action of rainfall in the wet season, and compounds rich in sugar and amino sugar are prone to preferential leaching and migration.

Excessive loss of SOM from the soil inevitably leads to soil nutrient deficiencies. Therefore, it is necessary to reduce farming and increase the area of trees and economic forests to combat physical erosion. At the same time, interplanting plants with shorter roots can preserve the organic carbon in the topsoil.

## 5. Conclusions

In this study, we collected soil from five vegetation restoration types in typical karst rocky desertification drainage on the basis of altitude. SOM was characterized by stable isotopes and fluorescence spectroscopy, and the key findings were as follows: Soil humification is weak and microbial activity is low in the Rocky desertification drainage. The changes of organic carbon content, δ^13^C_soc_, and various spectral indexes indicated that SOC in abandoned land was more enriched than in other vegetation restoration types, and the spectral properties of the soil were more stable, aromaticity was stronger, and molecular weight was larger. Thus, abandonment could be a more suitable control measure for enhancing SOC quality in the karst regions of south China. Three kinds of fluorescence components were detected: C1 and C2, which are exogenous humus-like, and C3, which is endogenous protein-like. The changes of organic carbon content, δ^13^C_soc_, various spectral indexes, and fluorescence components from A2 to A3 showed the change law of soil redistribution. The response of organic carbon erosion in the 0–15 cm soil layer was stronger than that in the 15–30 cm soil layer. Soil erosion preferentially destroyed aggregates and other structures in the upper soil, and unstable carbon was deposited in the lower part of the soil by runoff, soil flow, and gravity.

In rocky desertification area with serious soil erosion, SOM had an obvious response to soil erosion, especially in the 0–15 cm soil. Without considering the time cost, abandonment is more beneficial to soil stability. In addition, consideration can be given to maintaining shallow SOC by interplanting plants with developed and short roots. Therefore, for eroded soil, more attention should be paid to where organic carbon goes, and emphasizing carbon management rather than carbon storage.

## Figures and Tables

**Figure 1 ijerph-19-03210-f001:**
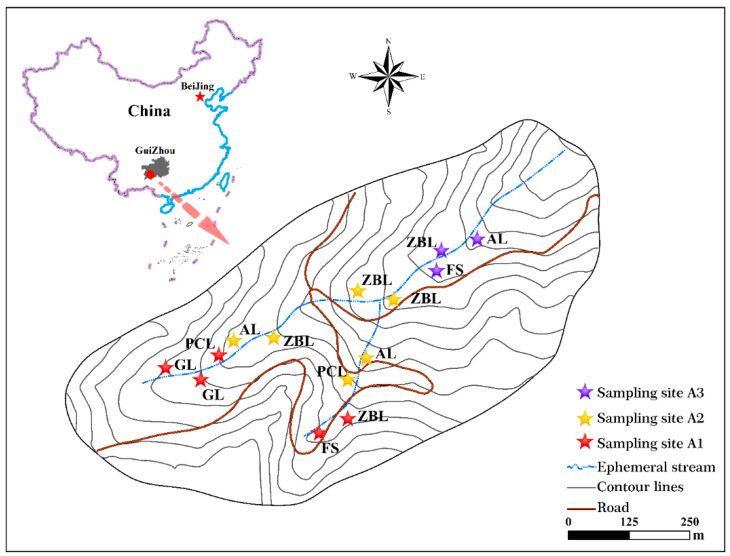
The location and sample point distribution of the drainage.

**Figure 2 ijerph-19-03210-f002:**
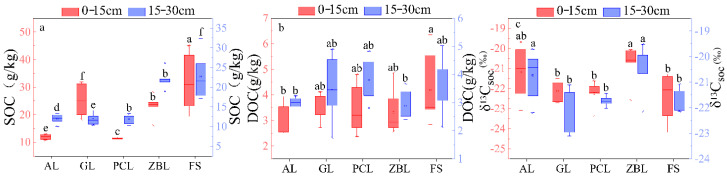
Box plots of soil SOC (**a**), DOC (**b**), and δ^13^C_soc_ (**c**) for 5 different vegetation restoration types. The boxes and the horizontal lines represent the mean and the median, respectively. The horizontal edge of the box represents 25% and 75% digits, the I shape line represents 10% and 90% digits, and the discrete points represent outliers. Different letters in the same soil layer indicate significant differences between land use types at *p* < 0.05.

**Figure 3 ijerph-19-03210-f003:**
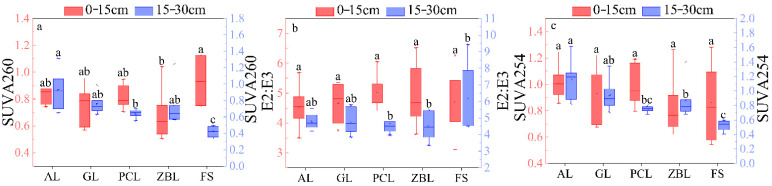
Box plots of soil SUVA254 (**c**), E2:E3 (**b**), and SUVA260 (**a**) for 5 different vegetation restoration types. The boxes and the horizontal lines represent the mean and the median, respectively. The horizontal edge of the box represents 25% and 75% digits, the I shape line represents 10% and 90% digits, and the discrete points represent outliers. Different letters in the same soil layer indicate significant differences between land use types at *p* < 0.05.

**Figure 4 ijerph-19-03210-f004:**
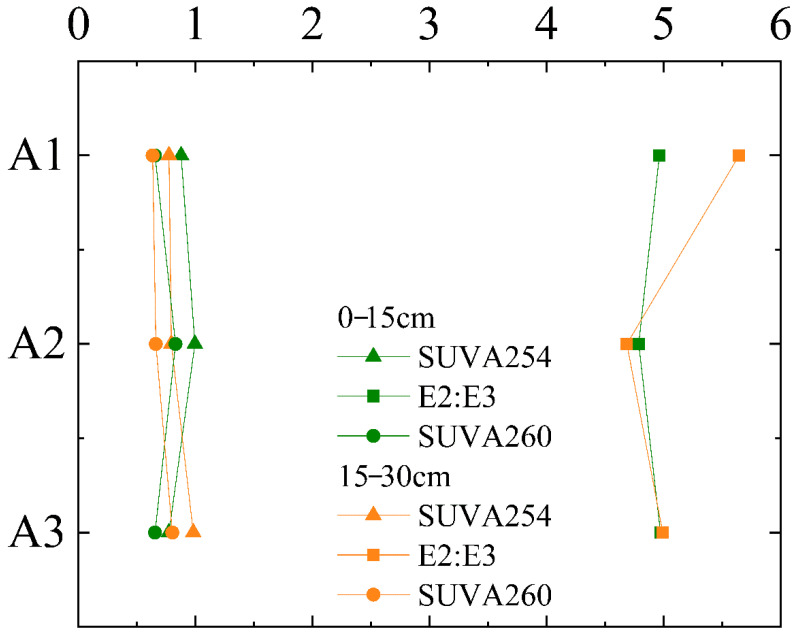
UV–visible absorption indexes of DOM derived from different sites and depths.

**Figure 5 ijerph-19-03210-f005:**
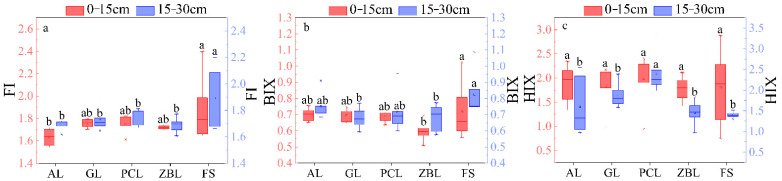
Box plots of soil HIX (**c**), FI (**a**), and BIX (**b**) for 5 different vegetation restoration types. The boxes and the horizontal lines represent the mean and the median, respectively. The horizontal edge of the box represents 25% and 75% digits, the I shape line represents 10% and 90% digits, and the discrete points represent outliers. Different letters in same soil layer indicate significant differences between land use types at *p* < 0.05.

**Figure 6 ijerph-19-03210-f006:**
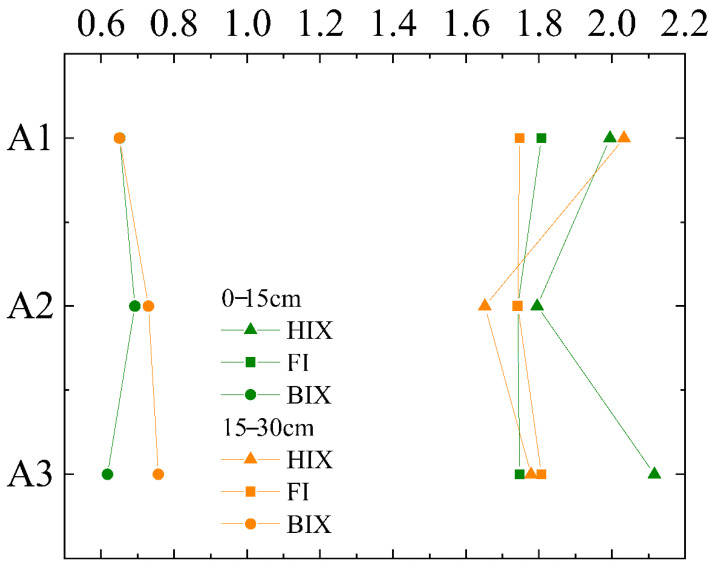
Distribution of fluorescence indexes of DOM in soil.

**Figure 7 ijerph-19-03210-f007:**
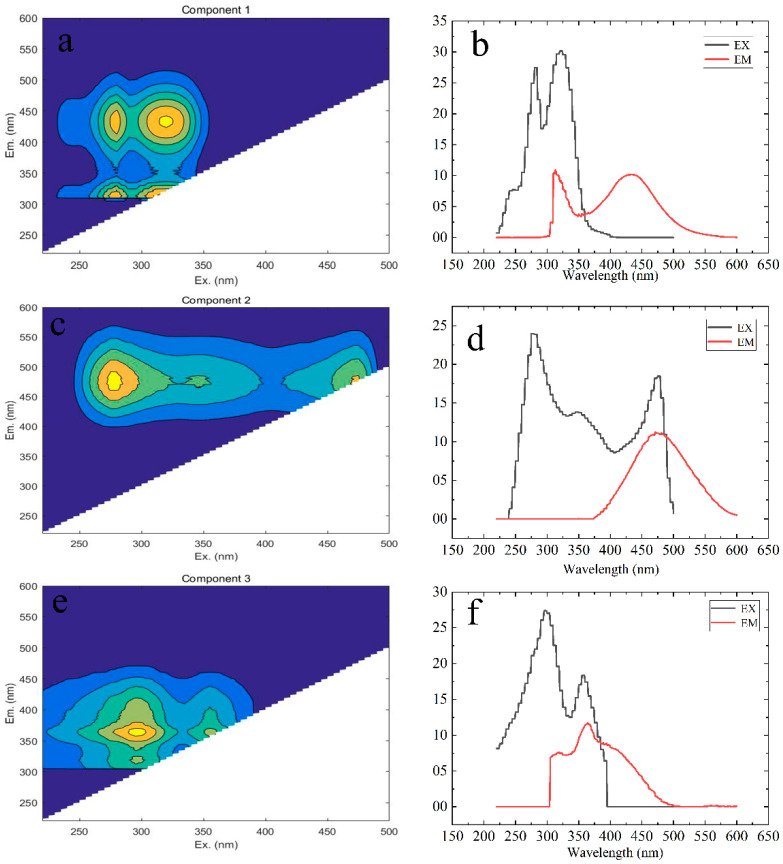
Fluorescence component diagram of DOM in soil. (**a**,**b**): intensify contour and split-half analysis of component C1; (**c**,**d**): intensify contour and split-half analysis of component C2; (**e**,**f**): intensify contour and split-half analysis of component C3.

**Figure 8 ijerph-19-03210-f008:**
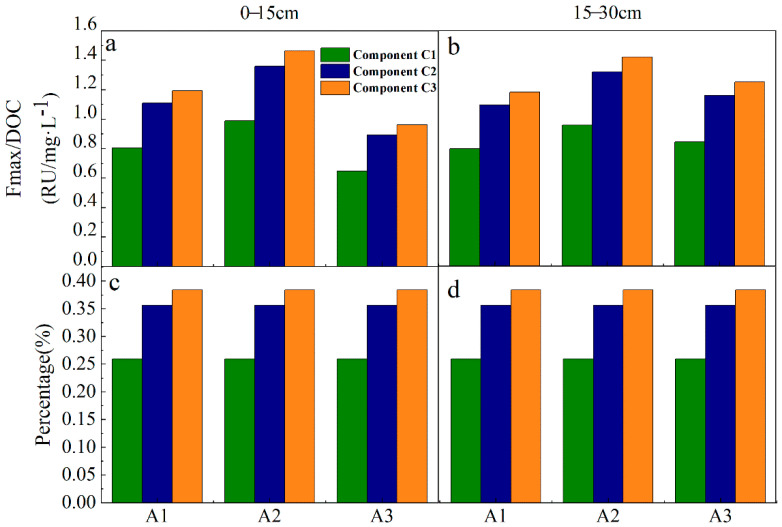
Distribution of three fluorescent components at different positions and depths. (**a**,**c**) The 0–15 cm soil layer; (**b**,**d**) the 15–30 cm soil layer.

**Table 1 ijerph-19-03210-t001:** Basic information of sampling points.

Site	Vegetation Restoration Type	Rock Outcrop Ratio	Slope	Vegetation Coverage
A1	GL	0%	3°	95%
A1	GL	5%	2°	95%
A1	FS	50%	5°	80%
A1	PCL	0%	5°	99%
A1	ZBL	50%	5°	50%
A2	AL	20%	3°	70%
A2	ZBL	50%	5°	50%
A2	PCL	15%	2°	30%
A2	GL	5%	2°	95%
A2	ZBL	10%	2°	40%
A2	ZBL	40%	2°	38%
A3	ZBL	40%	2°	38%
A3	GL	5%	2°	95%
A3	FS	55%	5°	75%

A1: upper reaches; A2: middle reaches; A3: lower reaches; AL: abandoned land; GL: grassland; PCL: peanut cultivated land; ZBL: *Zanthoxylum bungeanum* land; FS: forest.

**Table 2 ijerph-19-03210-t002:** The range of soil pH, SOC, DOC, and δ^13^C_soc_ in the drainage.

Sites	Depth (cm)	pH	SOC (g/kg)	DOC (mg/kg)	δ^13^C_soc_ (‰)
A1	0–15	7.30 ± 0.07 a	31.93 ± 1.73 a	96.95 ± 7.40 ab	−21.53 ± 0.28 b
A1	15–30	7.36 ± 0.07 a	27.06 ± 1.19 a	85.79 ± 7.20 ab	−21.97 ± 0.36 ab
A2	0–15	7.34 ± 0.07 a	28.27 ± 1.69 a	79.36 ± 4.13 b	−21.06 ± 0.23 b
A2	15–30	7.41 ± 0.05 a	25.35 ± 1.37 a	75.88 ± 3.06 a	−21.79 ± 0.27 a
A3	0–15	7.46 ± 0.09 a	27.73 ± 3.05 b	125.99 ± 13.29 a	−22.44 ± 0.30 b
A3	15–30	7.52 ± 0.09 a	26.09 ± 2.82 b	89.76 ± 7.83 a	−22.64 ± 0.29 b

Different letters in the same soil layer indicate significant differences between land use types at *p* < 0.05. SOC: soil organic carbon; DOC: dissolved organic carbon.

## Data Availability

The data presented in this study are available on request from the corresponding author.

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
