# Peer review of "Soil Organic Carbon Distribution and Its Response to Soil Erosion Based on EEM-PARAFAC and Stable Carbon Isotope, a Field Study in the Rocky Desertification Control of South China Karst"

_ijerph, 2022, doi:10.3390/ijerph19063210_

Round 1

Reviewer 1 Report

It is interesting to evaluate the ecological restoration of rock desertification in karst area. However, the background is not well documented. The hypothesis proposed here are no sensing that could be expected in normal situation. There is a lack of related demonstration to propose the  hypothesis. According to the results, the evidences are weak in supporting the conclusions. SOC is a concentration of intensity not of capacity. Thus, soil bulk density is needed to calculate organic carbon sequestration in soil. Moreover, it is difficult to explain the conclusion of abandoned soil was more obvious than other vegetation restoration in organic carbon enrichment. There are too many flaws in the manuscript.

Reviewer 2 Report

This is an interesting study and the authors have collected a good dataset. I recommend this manuscript for publication in MDPI International Journal of Environmental Research and Public Health after Minor revisions, given below:

Abstract

The justification of the study is not well demonstrated.

L11-13; the authors may need to briefly mention soil erosion in the study site prior to this sentence. Soil erosion has for a long time been regarded as a process leading to soil organic matter (SOM) loss. So, I would raise the issue of soil erosion and a gap of knowledge, and the aim.

Authors may need first to define SOC, DOC, DOM, and PARAFAC, and then do the abbreviation.

Introduction

L62-65: these sentences do not seem related to the topic. Authors need to clearly illustrate the study aim and result implications here.

L70-73: I did not see any logic of the hypotheses especially hypothesis 1. In lines 60-62, authors mentioned increase in vegetation cover in the site, however they need to argue whether vegetation restoration leads to soil organic carbon accumulation.

Materials and Methods

Please fix the orders. In Line 109, I see “2.2.1”, but I did not see “2.2.2”.

L173: “Statistical analysis” you may need to change the order to “2.5”

Results and Analysis

No comments

Discussion

No comments

Concussions

No comments.

Reviewer 3 Report

1) This paper appears to be an interesting contribution to research on ‘Soil organic carbon distribution and its response to soil erosion’ ‘in karst areas’;

2) Its academic and scientific values make it suitable for the open access, peer-reviewed International Journal of Environmental Research and Public Health (ISSN 1660-4601);

3) However, there are some minor spell checking, formatting and writing issues that must be addressed by the authors. See, for instance, the “examples” listed below (underlined text):

  • Line 7, CHECK ENGLISH: ‘State Engineering Technology Institute for Karst Desertfication? Control’;
  • Line 26, CHECK for a possible ‘EXTRA PERIOD’: ‘soil organic carbon; vegetation restoration; soil erosion; dissolved organic matter.? stable carbon isotope; karst
  • Lines 53-54, CHECK TYPO: ‘carbon sinks in karst regions in China amount to 36.991 million tons of CO2/a [16]’;
  • Lines 79-80, CHECK for a possible ‘EXTRA PERIOD’: ‘Rock desertification is the most serious ecological and environmental problem facing this region.? [19].’;
  • Line 152, CHECK TEXT: ‘2.4.δ13. Csoc and spectral index’;
  • Line 153, CHECK ‘Formula for calculating δ13Csoc isotope ratio of samples’ for ‘non-English/unreadable symbols’;
  • Lines 156-158, CHECK FORMULA (2) and TEXT for CORRECTNESS: ‘a(λ) is the absorption coefficient at wavelength λ, a(λ) is the absorbance when wavelength changes, and r is the optical path length (r=0.01m).’;
  • Line 173, CHECK SECTION NUMBERING: ‘2.4. Statistical analysis’;
  • Lines 185-186, CHECK TYPO: ‘. In 0-15cm soil layer, DOC concent? at site A3 was 24.19% higher’;
  • Lines 187-188, CHEC TYPO: ‘and there was no significant difference in DOC concent in 15-30 cm soil layer’;
  • Line 208, CHECK ENGLISH: ‘the distribution of UV-Vsible absorption indexs’;
  • Line 224, Figure 4, CAPTION, CHECK ENGLISH: ‘UV-Vsible absorption indexs of DOM derived from different sites and depths’;
  • Lines 235-236, CHECK WORDING: ‘BIX values were mostly below 0.8 (Figure 6), it showed that weak soil activity [32].’;
  • Lines 248-251, CHECK WORDING: ‘EEM-PARAFAC model was used to identify DOM fluorescence components of soil in the drainage, and three fluorescence components were obtained (Figure 7), C1(Ex/Em=320/ 434), C2 (Ex/Em =290(360)/ 485) can be classified as exogenous humic-like [7、35], C3 (Ex/Em =280/ 318) can be classified as endogenous protein-like [36].’;
  • Lines 274-275, CHECK ENGLISH: ‘There was?? no significant differences in fluorescence indexs between different vegetation restoration types (Figure 5),’;
  • Lines 299-301, CHECK ENGLISH: ‘Due to the influence of periodically flooded and scour, organic matter of soil and sediment in the water-level fluctuation zone will be released into the water [45].’;
  • Lines 307-308, CHECK WORDING: ‘In this study, SUVA254 decreased and BIX value decreased from site A2 to site A3, it showed that site A3 was less aromaticity and more exogenous.’;
  • Line 400, CHECK TYPO: ‘16. Jiang, Z.; Qin, X.; Cao, J.; Jiang, X.; He, S.; Luo, W. 2011. Zonal calculation of atmospheric CO 2 carbon sink…’.

Reviewer 4 Report

This article presents an investigation on soil organic carbon distribution and its response to soil erosion 2 based on EEM-PARAFAC and stable carbon isotope in South China Karst. At a glance, this manuscript indicates the rigorousness of the research presented. However, some missing information and inadequate presentations require significant revisions/additions. The following comments intend to enable the authors to disseminate their work at the highest possible quality.

  1. Introduction.
    • Throughout the section. Rather than directly putting reference numbers at the end of sentences, the authors should add statements on the references (what they were talking about) in support of the previous (original) statement. At least one statement is required to state the idea lifted from each reference.
    • Lines 60-62. Despite stating "previous studies," the authors fail to mention what those studies have done. Add direct references to those studies. Each study would require at least one statement to elaborate the ideas.
    • Lines 64-65. The authors should elaborate more on the "few ways." Add relevant references as necessary, and explain their coverage separately.
  2. Materials and Methods.
    • Lines 76-96. These lines require more references to support the data and information. Every data provided and trend mentioned must be based on accessible literature/archives. Therefore, the authors must cite those literature/archives properly.
    • Line 153. Some indices in Equation 1 are in Chinese characters. The authors should consistently use roman/greek alphabets to explain any formulas.
    • Table 1. The authors should put table notes to explain the abbreviations. It would make the table self-explanatory.
  3. Results and Analysis.
    • Table 2. The authors should distinguish "Sites" codification and "Depth" data into two separate columns. Do not forget to use square brackets for units of measurements.
    • Table 2. Please refer to the comment on Table 1 for table notes.
  4. Conclusion.
    • The current conclusion is not adequate to conclude the entire research. The authors should deliver a well-balanced portion between the following three core parts. Consider making one paragraph for each part. 
      1. Part 1: Summary of this study.
      2. Part 2: Present key findings.
      3. Part 3: Suggest managerial implications (beyond engineering) based on the key findings of this study.
    • NOTE: Existing paragraphs in this section can be merged/included in Part 2.

Round 2

Reviewer 1 Report

The revision was improved. Language should be still improved. Minor revision is needed before acceptance.

L85, 55*104 km2, where 4 should be superscripted.

L105, The title of figure 1 should be modified.

L243, 0-15cm soil layer, ..... is not a complete sentence.

L248. what is soil activity?

L284, soil covered by different..., these maybe lead to ...Not supported.

L303, 15-30 cm soil layer had --> the soil in 15-30 layer had

L497, reference 47 is blank.

Reviewer 4 Report

After a thorough check on the revised manuscript, I see that the authors have put effort into revising their first submission. I want to suggest the following revisions for this second review round.

Introduction.

  • The hypotheses should be the product of a rigorous thought process by referring to relevant literature. For each hypothesis, the authors should elaborate on it by providing arguments leading to the reasons why they end up suggesting the hypothesis. Please cite relevant references to support each argument.

Table 2.

  • Again, please use square brackets "[]" instead of round brackets "()" to indicate units of measurements (e.g., "[g/kg]" instead of "(g/kg)"). Please be noted that percentage is a unitless measurement, so in that case, round brackets are correct. Then, please move the unit in column 2 (Depth) to the heading row (i.e., "Depth [cm]").
  • NOTE: Please also ensure the consistent use of square brackets for units of measurements throughout the article.

Conclusion.

  • The discussion centers on two issues ("Spectral characteristics and isotopic composition of soils of different vegetation restoration types" and "Effects of soil erosion on the distribution of soil organic matter in the drainage"). Consequently, the authors should elaborate managerial implications on those two issues as references for relevant stakeholders. It would help argue the usefulness of this research beyond engineering matters.

I appreciate all efforts the authors have made to address the concerns of the reviewers. I would like to say a piece of good luck with the publication, and for the continuity of research on similar or other relevant topics.
